# Signal-Independent Background Calibration with Fast Convergence Speed in Pipeline-SAR ADC

**DOI:** 10.3390/mi14020300

**Published:** 2023-01-23

**Authors:** Yu-Jun Wang, Peng Wang, Li-Xi Wan, Zhi Jin

**Affiliations:** 1Institute of Microelectronics of Chinese Academy of Sciences, Beijing 100029, China; 2University of Chinese Academy of Sciences, Beijing 100049, China; 3Chengdu Tiger Microelectronics Research Institute Co., Ltd., Chengdu 610000, China; 4School of Integrated Circuits, Tsinghua University, Beijing 611731, China

**Keywords:** noise background calibration, signal independence, fast convergence, pipeline-SAR ADC

## Abstract

This brief proposes a signal-independent background calibration in pipeline-SAR analog-to-digital converters (ADCs) with a convergence-accelerated technique. To achieve signal independence, an auxiliary capacitor array C_A_ is introduced to pre-inject a pseudo-random noise (PN) in the sampling phase to cancel out the opposite PN injection of the calibrated capacitor in the conversion phase, and C_A_ is also used to realize the D/A function of the calibrated capacitor in the conversion phase. In this way, no matter what the signal is, the residue headroom remains unchanged even with PN injection. Moreover, the first sub-ADC is designed with extended conversion bits to quantize its own residue after delivering the conversion bits required by the first stage. Afterwards, this result is provided to the calibration algorithm to reduce the signal component and accelerate the convergence. Based on the simulation, the signal-to-noise and distortion ratio (SNDR) and spur-free dynamic range (SFDR) improve from 45.3 dB and 56.4 dB to 68.2 dB and 88.4 dB, respectively, after calibration. In addition, with the acceleration technique, convergence cycles decrease from 1.7 × 10^8^ to 5.8 × 10^6^. Moreover, no matter whether the input signal is DC, sine wave or band-limited white noise, the calibration all works normally.

## 1. Introduction

The successive approximation register analogue-to-digital converter (SAR ADC) is the most power- and area-efficient architecture to attain 8~12-bit resolution [1,2], while its conversion speed is limited due to the inherent serial conversion process. The pipeline-SAR ADC splits the SAR ADC into two stages and connects them with an amplifier to realize pipelined operation, not only reducing conversion time, but also suppressing the comparator noise [3,4]. However, the induced inter-stage gain error and inevitable capacitor mismatch error cause bit-weight errors and, thus, degrade ADC performance. Foreground calibration can extract those errors and compensate them in the digital or analog domain, but cannot track environment variation and thereby requires a high-gain amplifier [5], such as the ring amplifier [6] or telescopic amplifier, which are power-hungry and have complex structures. Background calibration can track the gain variation of the amplifier and allows employing a power-efficient open-loop amplifier, such as the dynamic amplifier [7,8] or gm-R amplifier [9] in pipeline-SAR ADCs. The signal-dependent background calibration is facilitated by detecting the signal value and injecting PNs correlated with the bit-weight errors into the residue path [10]. The correct bit weights can be achieved by exploiting the least mean square (LMS) iterative algorithm, which only involves addition and shift operations. Nevertheless, the signal dependence causes failure of the calibration when the input signal is DC or has a small dynamic range. In [11], the foreground and signal-independent background calibration are adopted to treat capacitor mismatch and inter-stage gain error, respectively, but the demand of multi-bit multiplication and division operation to update bit weights makes the method complex. Moreover, the signal independence increases the signal interference and, thus, decreases the algorithm’s convergence speed. The split-ADC calibration [12], where the ADC is divided into two identical half-ADCs to convert the input signal into different paths, exploits the outputs’ difference to eliminate the signal interference, and, therefore, increases the convergence speed, although it incurs more area and power consumption since the parasitic effect and logic circuit almost double. Sharing the first stage with two second stages in [13] achieves the same acceleration effect as split-ADC, but the residue requires two amplifications, which reduces the ADC speed.

In this brief, a signal-independent background calibration with short convergence time is implemented by introducing an auxiliary capacitor array and modifying the sub-ADC structure. To realize signal independence, PN is pre-injected into an auxiliary capacitor array C_A_ in the sampling phase to counteract the opposite PN injection of the calibrated capacitor in the conversion phase, and C_A_ is also used to realize the D/A function of the calibrated capacitor in the conversion phase. Using a coarse sub-ADC to convert the first stage’s result into pipeline-SAR ADC can not only improve speed and power, but also loosen the reference voltage settling [14,15]. The calibration modifies the sub-ADC with extended conversion bits to quantize its own residue after delivering the conversion bits required by the first stage. Afterwards, this result is provided to the calibration algorithm to reduce signal interference and accelerate the convergence.

The brief is organized as follows: Section 2 introduces the proposed calibration in detail. The simulation results and comparison are presented in Section 3. Section 4 concludes the brief.

## 2. Proposed Background Calibration

The block diagram of the proposed pipeline-SAR ADC is shown in Figure 1a with single-end structure, while the actual design is differential. The two stages are configured as 5 bits and 8 bits with 1-bit redundancy, and the inter-stage gain is 8 instead of 16 for reducing the amplifier output swing. The four most significant bits (MSBs) of the 1st stage are thermometer-coded and the least significant bit (LSB) is binary, and they comprise the 1st stage main DAC for generating the residue after SAR conversion. An auxiliary capacitor array C_A_ whose value is equal to one MSB unit capacitor is introduced to the 1st stage CDAC for working alternatively with the main DAC’s capacitor that PN is injected into. The values of the 1st stage CADC are depicted in Figure 1a, in which C_M15_ߝC_M0_ and C_A3_–C_A0_ represent the main DAC capacitors and the auxiliary capacitors, respectively, while C_5_–C_1_ refer to capacitors corresponding to the 5 bits of the 1st stage. An 8-bit sub-ADC is implemented and configured as 5 + 4 bits with 1-bit redundancy, and the 5 bits are delivered to the 1st stage CDAC for generating residue while the 4 bits are transmitted to the calibration engine for speeding up calibration. The timing diagram is illustrated in Figure 1b. The sub-ADC attains the 4-bit conversion results used for acceleration when the 1st stage’s residue is amplified, which does not reduce ADC speed. Since the noise in sub-ADC does not affect the main ADC’s outputs for its redundancy, the sampling capacitors occupy a small area and the comparator is low-power. Nonetheless, the residue of the 1st stage and sub-ADC after 5 MSBs’ conversion are not equal due to the capacitors’ mismatch, and will limit the signal elimination effect. In Section 3, Monte Carlo simulations are performed to evaluate the negative effect.

The simplified signal flow diagram is illustrated in Figure 1c. Firstly, the input signal V_in_ is sampled by both the 1st stage and sub-ADC as in [14]. Then, the sub-ADC converts V_in_ into 5-bit digital results, D_M_, that are subtracted from V_in_ in the 1st stage and sub-ADC simultaneously to obtain the residue voltage V_RES1_ and V_RESA_, respectively. Furthermore, PN is added to the 1st stage’s V_IN_ for extracting inter-stage gain [10]. Afterwards, V_RES1_ is amplified G_R_ times and the amplified voltage V_AMP_ is converted by the 2nd stage into 8-bit D_2_. Meanwhile, the residue V_RESA_ is converted into 4-bit D_L_ by sub-ADC. After that, D_M_, D_L_ and D_2_ are passed to the calibration engine to attain the calibrated gain G_c_. Finally, the digital part utilizes D_M_, D_L_ and G_C_ to achieve the 12-bit output D_out_.

### 2.1. Signal-Independent PN Injection

For extracting the bit weights, PN should be correlated with the capacitors of the 1st stage SAR DAC (C_5_-C_1_). However, C_5_-C_1_ should be connected to D_M5_-D_M1_, the 5 MSBs’ conversion results of sub-ADC, to generate the 1st stage residue. To resolve this problem, the auxiliary capacitor C_A_ is introduced to replace the injected part to generate residue.

C_5_-C_2_ change to thermometer code C_M15_-C_M1_ so that C_A_ can be 2C_u_ instead of 16C_u_ to save area. Nevertheless, the residue also increases due to the injection and saturates the 2nd stage. An opposite PN is pre-injected into C_A_ to cancel out residue change. Consequently, PN can be injected into any one of C_M15_-C_M1_ whatever the signal is. Taking C_M1_ as an example, Figure 2 shows how PN is injected into C_M1_. During the sampling phase, PN is sampled on the bottom plate of C_A3_-C_A0_, and V_in_ is sampled on the bottom plate of C_M15_-C_M0_. After sub-ADC finishes 5 MSBs’ conversion, the residue-generating phase comes, and C_A3_-C_A0_ are connected to D_M2_ while C_M1_ is connected to PN. The residue V_res_ on the top plate can be derived as
(1)$$V_{res}=V_{in}-(\sum_{i=1}^{5}D_{Mi}C_{i}+D_{M2}(C_{M1}-C_{A})-PN(C_{M1}-C_{A}))\frac{V_{ref}}{C_{t}},$$
where V_ref_ denotes the reference voltage and C_t_ refers to the total value of C_M15_-C_M0_. Equation (1) shows PN is injected into C_M1_-C_A_. If there is no mismatch between C_M2_ and C_A_, the residue is
(2)$$V_{res}=V_{in}-\sum_{i=1}^{5}D_{Mi}C_{i}\frac{V_{ref}}{C_{t}},$$
which indicates that PN injection is canceled out and the residue remains unchanged.

In fact, only the values of C_M1_-C_A_ can be obtained after calibration, and C_A_ should be measured first. C_A_ is divided into C_A3_~C_A0_ for reducing the residue range overhead during PN injection. Table 1 shows the detailed PN injection configurations, where only C_M1_ is listed because C_M15_-C_M1_ are similar. PNs are injected into C_A0_, C_A1_-C_A0_, C_A2_-C_A1_-C_A0_, C_A3_-C_A2_-C_A1_-C_A0_, C_M1_-C_A3_, C_M0_-C_A_, respectively, and C_A3_~C_A0_ and C_M1_~C_M0_ are obtained by adding those delta values. The maximum residue range is [−1.25, 1.25] LSB when PN is injected into C_A0_, and there is still a 0.75-bit redundancy left of the 2nd stage quantization range.

### 2.2. Signal Elimination and Convergence Acceleration

The residue with PN is amplified and converted by the 2nd stage, and, taking PN injected into C_M2_ as an example, we can obtain
(3)$$\begin{array}{ll} D_{BE}= & G_{R}(V_{in}-\sum_{i=1}^{5}D_{Mi}\frac{C_{i}}{C_{t}}V_{ref})-D_{M2}(BW_{M2}-BW_{A})\\ & +PN(BW_{M2}-BW_{A}) \end{array}$$
where D_BE_ is the 2nd stage conversion result, G_R_ is the real inter-stage gain and BW_X_ refers to the bit weight corresponding to C_X_. There are three items on the right side of (3), but only the third one is correlated with PN, and the two PN-uncorrelated items are defined as signal interference V_itf_, which is given by
(4)$$V_{itf}=G_{R}(V_{in}-\sum_{i=1}^{5}D_{Mi}\frac{C_{i}}{C_{t}}V_{ref})-D_{M2}(BW_{M2}-BW_{A}).$$

In the traditional calibration, by multiplying D_BE_ with PN to remove uncorrelated items and exploiting the product in the LMS algorithm, BW_M2_-BW_A_ is achieved as follows:(5)$$\begin{array}{l} \Delta BW_{M2}(n+1)=\Delta BW_{M2}(n+1)-u\times (\Delta BW_{M2}(n)-D_{BE}(n)\times PN)\\ \;\Delta BW_{M2}=BW_{M2}-BW_{A} \end{array},$$
where μ denotes the learning step. Figure 3 shows the learning curve of $\Delta BW_{M2}$ including two phases, track phase and stable phase. In the track phase, since the average value of PN× V_itf_ is 0, μ is the only factor that determines the convergence speed. During the stable phase, $\Delta BW_{M2}$ fluctuates and the fluctuation’s amplitude depends on μ × V_itf_, which deteriorates the ADC performance. Therefore, to not degrade the ADC performance and accelerate convergence, μ × V_itf_ should decrease while u increases, which requires V_itf_ during the stable phase to be removed. The 2nd part of V_itf_ in (4) can be easily obtained, whereas the 1st part, which is the product of G_R_ and the residue of V_in_ after 5 conversions, is not available. Sub-ADC is used to quantize the residue, and we can obtain
(6)$$V_{in}-\sum_{i=1}^{5}D_{Mi}\frac{C_{i}}{C_{t}}V_{ref}+V_{off}=D_{s},$$
where Ds refers to the sub-ADC quantization result and V_off_ denotes the offset between sub-ADC and main ADC. To eliminate V_off_, each PN is injected twice in the two adjacent conversion cycles with the opposite symbol, and then (1 − z^−1^)/2 is adopted to filter D_S_. With the high pass filter, (6) can be rewritten as:(7)$$V_{in}-\sum_{i=1}^{5}D_{Mi}\frac{C_{i}}{C_{t}}V_{ref}=D_{s}.$$

In addition, since the real bit weight and the inter-stage gain change the same proportion compared to the ideal ones, the real inter-stage gain G_R_ can be estimated as
(8)$$G_{R}=\frac{BW_{5R}}{BW_{5I}}G_{I},$$
where BW_5I_ and BW_5R_ represent the ideal and real bit weight of the 1st stage MSB, and G_I_ is the ideal inter stage gain. BW_5R_ is replaced by BW_5(n)_ provided by the calibration, and they are only nearly equal in the LMS stable phase, which satisfies the requirement that only the V_tif_ needs to be eliminated during this phase. BW_5I_ and G_I_ are 2^11^ and 2^3^, hence (6) can be realized by 8-bit left shift of BW_5R_. Finally, the V_itf_ is estimated as
(9)$$V_{itf}(n)=BW_{5}(n)D_{s}(n)/2^{8}$$

Subtracting V_tif_ from D_BE_ can effectively suppress the signal interference and hence boost the calibration speed.

## 3. Simulation Results and Comparison

The proposed 12-bit pipeline-SAR ADC is simulated in MATLAB with 1V reference. The unit capacitor’s mismatch standard deviation σ of the first stage, sub-ADC 5 MSBs, second stage and sub-ADC 4 LSBs are 1%, 1%, 5% and 5%, respectively. Furthermore, the inter-stage gain is 7.2 with 20% error, compared with 8. In addition, σ of the offset between the sub-ADC and the main ADC is 5 mV, and the sampling capacitor of the first stage is 600 fF to add KT/C noise to signal.

To demonstrate the effectiveness of the proposed calibration algorithm, the full-scale sine wave input is simulated to observe the dynamic performance. One of the output spectra without and with calibration is shown in Figure 4. After calibration, SNDR and SFDR improve from 45.3 dB and 56.4 dB to 68.2 dB and 88.4 dB, respectively. Without and with signal elimination, simulations are executed with the same mismatch, and the dynamic performance learning curves are shown in Figure 5.

Although SNDR and SFDR are all over 67 dB and 80 dB after convergence, the convergence conversion cycles are 1.7 × 10^8^ and 5.8 × 10^6^ for without and with signal elimination. With DC, sine wave and the band-limited white noise input, the learning curves of bit weight BW5 are illustrated in Figure 6. Whatever the input is, BW5 can converge at about 6 × 10^6^ times and proves the signal independence.

To evaluate the effect of the sub-ADC 5 MSBs’ capacitor mismatch to the signal cancellation, 400-time Monte Carlo simulations are performed with σ = 1% unit capacitor mismatch. The output spectra after convergence (5.8 × 10^6^ cycles) are statistically analyzed. Figure 7 shows the SFDR distribution histogram with the mean value E of 86.3 dB and standard variation σ of 1.1 dB, while SNDR is 68 dB and hardly changes, thus, it is not shown in Figure 7. The unit capacitor’s mismatch changes from 1% to 5%, and each case is simulated 1000 times. Figure 8 shows the worst case in 3σ principle, i.e., E−3σ, of SFDR. With 1% mismatch, SFDR is 83.0 dB, while it is 78.0 dB with 5% mismatch, which still maintains good calibration effect. A 5% mismatch is easily achieved in most processes, and, thus, the mismatch of the sub-ADC’s 5 MSBs does not limit the convergence acceleration.

As shown in Table 2, state-of-the-art background calibrations are listed. Compared to [11,12], this work and [13] are signal-independent. To achieve signal independence, ref. [13] requires complex shuffling and two-bit inter-stage redundancy, while, in this work, an auxiliary capacitor array with PN pre-injection technique realizes simple shuffling and occupies little residue headroom. The convergence cycles are less than [11], but more than [12,13]. Although the signal elimination in the proposed calibration can speed up convergence, there is mismatch between sub-ADC and main ADC, and this means that the signal cannot be estimated as accurately as in [12,13]. However, split-ADC and double second stage introduce more circuit hardware overhead.

## 4. Conclusions

In this brief, a signal-independent calibration with a convergence acceleration technique is developed. By introducing an auxiliary capacitor array to replace the DAC injected with PN to generate residue, and to reduce the residue range with the PN pre-injection technique, the signal independence is implemented. Furthermore, the signal interference is cancelled with the sub-ADC’s quantization result, and hence the convergence cycles decrease. The calibration is applied in a 12-bit pipeline-SAR ADC, which improves the SNDR and SFDR by 22.9 dB and 33.2 dB, respectively. Moreover, calibration converges in 5.8 × 10^6^ instead of 1.7 × 10^8^ cycles with signal elimination. In addition, the calibration operates normally no matter whether the input signal is DC, sine wave or band-limited white noise.

## Figures and Tables

**Figure 1 micromachines-14-00300-f001:**
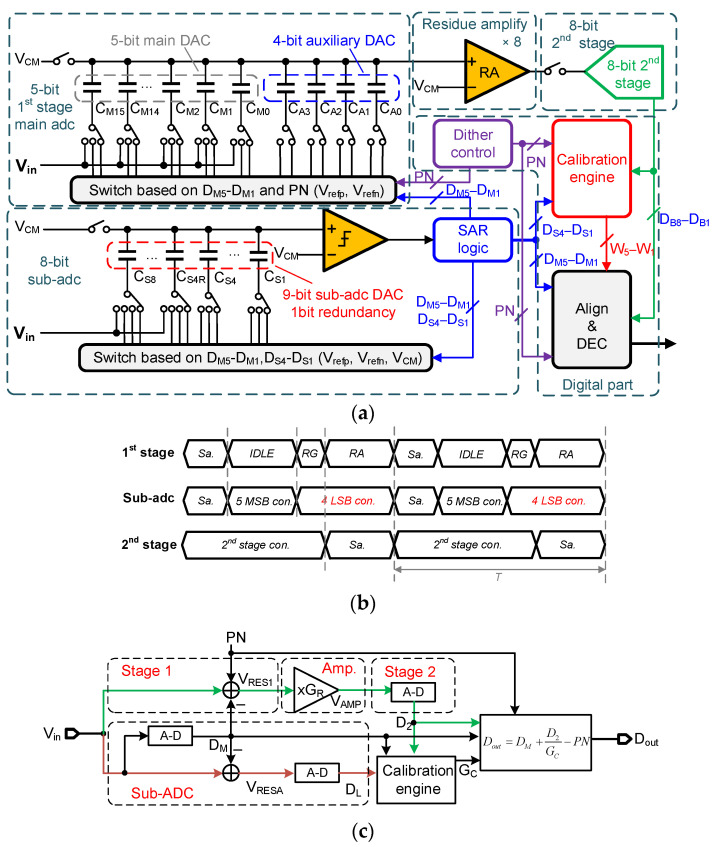
(**a**) Block diagram, (**b**) timing diagram and (**c**) simplified signal flow diagram of the proposed pipeline-SAR ADC.

**Figure 2 micromachines-14-00300-f002:**
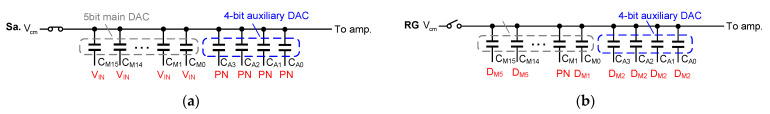
The connection configurations of the 1st stage CDAC when PN is injected into C_M1_ (**a**) in sampling phase; (**b**) in residue generation phase.

**Figure 3 micromachines-14-00300-f003:**
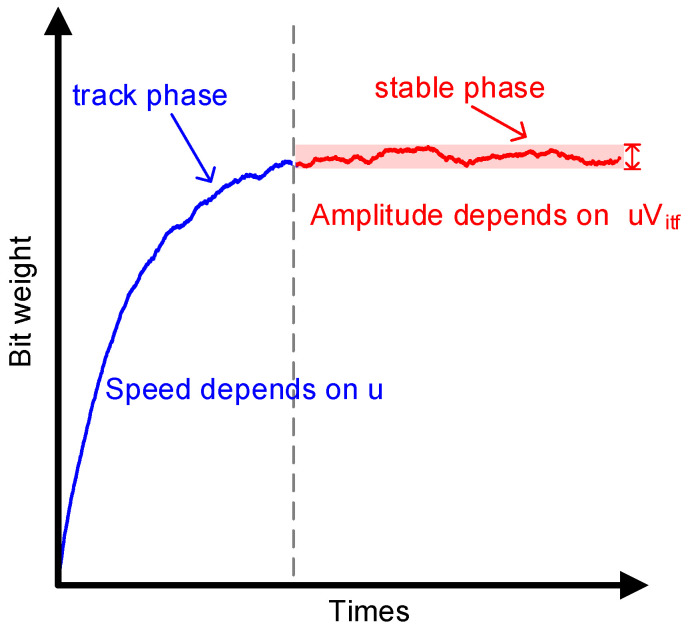
Bit-weight learning curve.

**Figure 4 micromachines-14-00300-f004:**
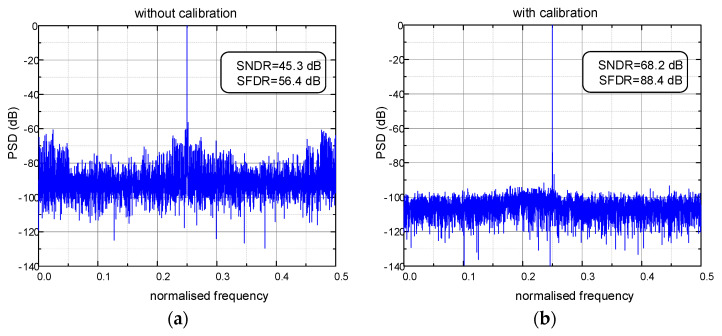
Output spectra: (**a**) without calibration; (**b**) with calibration.

**Figure 5 micromachines-14-00300-f005:**
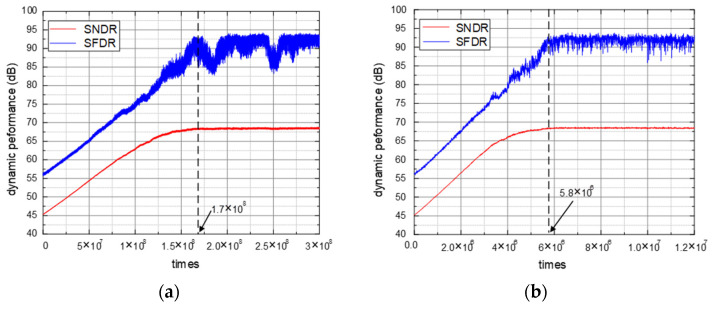
ADC dynamic performance learning curves: (**a**) without signal elimination; (**b**) with signal elimination.

**Figure 6 micromachines-14-00300-f006:**
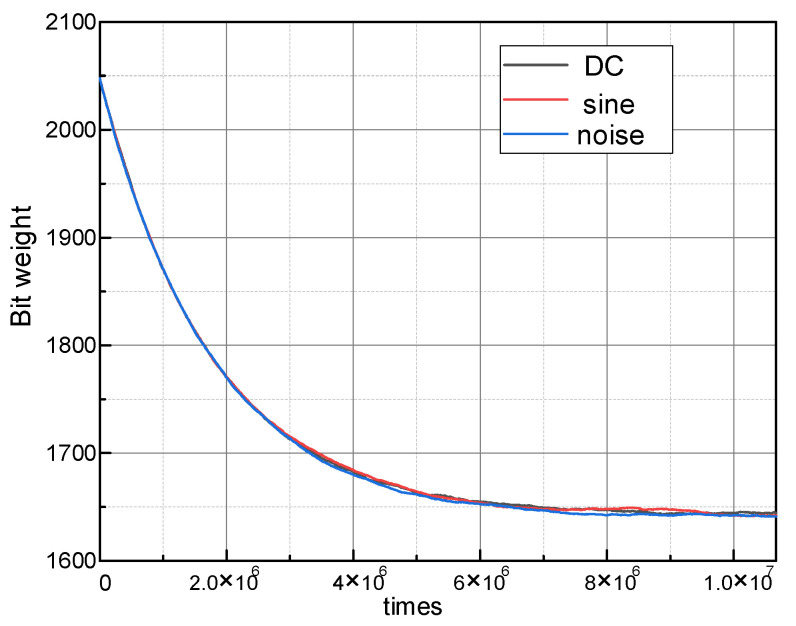
MSB learning curve with DC, square and band-limited white noise inputs.

**Figure 7 micromachines-14-00300-f007:**
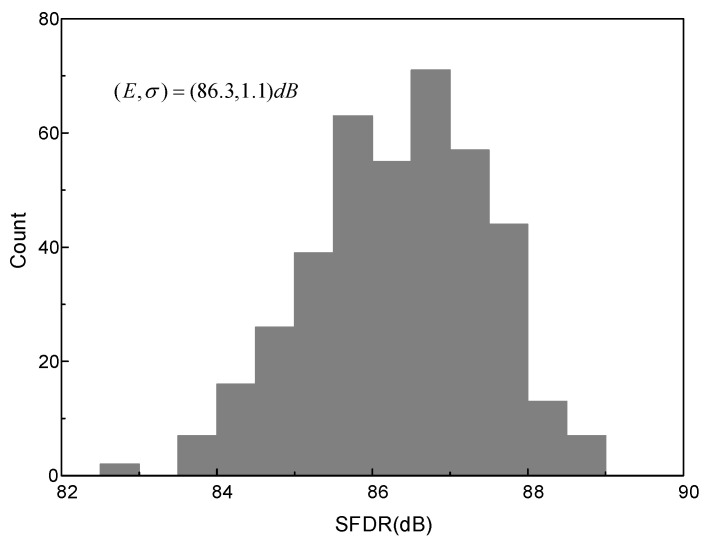
Histogram of output spectra’s SFDR with mismatch of 1% of sub-ADC 5 MSBs.

**Figure 8 micromachines-14-00300-f008:**
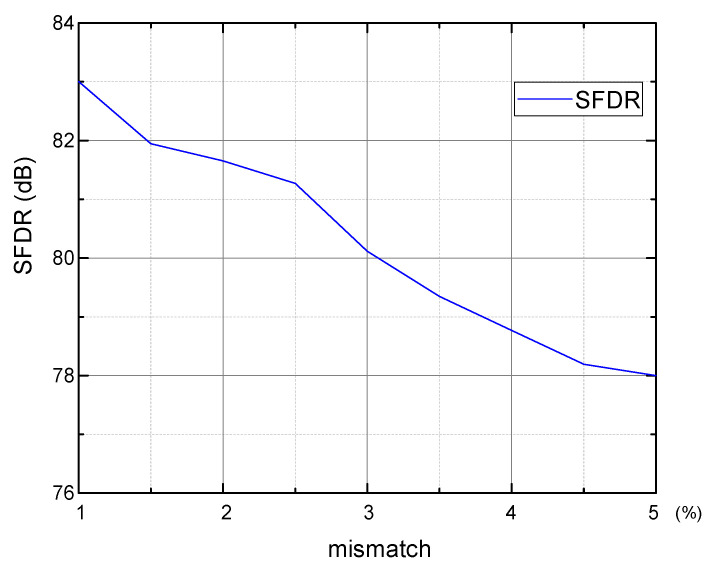
SFDR with different mismatch of sub-ADC 5 MSBs.

**Table 1 micromachines-14-00300-t001:** PN Injection in the First Stage CDAC.

Inj. Cap	Connection of (C_M1_, C_M0_, C_A3_, C_A2_, C_A1_, C_A0_)	
	Sampling Phase	Residue-Generation Phase
C_A0_	(V_in_, V_in_, 0, 0, 0, −PN)	(D_M2_, D_M1_, 0, 0, 0, 0)
C_A1_	(V_in_, V_in_, 0, 0, −PN, PN)	(D_M2_, D_M1_, 0, 0, 0, 0)
C_A2_	(V_in_, V_in_, 0, −PN, PN, PN)	(D_M2_, D_M1_, 0, 0, 0, 0)
C_A3_	(V_in_, V_in_, PN, −PN, PN, PN)	(D_M2_, D_M1_, 0, 0, 0, 0)
C_M0_	(V_in_, V_in_, PN, PN, PN, PN)	(D_M2_, PN, D_M1_, 0, 0, 0)
C_M1_	(V_in_, V_in_, PN, PN, PN, PN)	(PN, D_M1_, D_M2_, D_M2_, D_M2_, D_M2_)

**Table 2 micromachines-14-00300-t002:** Comparison with state-of-the-art calibrations.

	[11]	[12]	[13]	This Work
Signal-indep.	No	No	Yes	Yes
Architecture	Traditional	Split-ADC	Double 2nd stage	Sub-ADC
Converg. cycles	2.6 × 10^7^	5 × 10^6^	10^6^	5.8 × 10^6^
Extend period	No	No	Yes	No

## Data Availability

Not applicable.

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
