# Peer review of "Signal-Independent Background Calibration with Fast Convergence Speed in Pipeline-SAR ADC"

_micromachines, 2023, doi:10.3390/mi14020300_

Round 1

Reviewer 1 Report

The paper is interesting. However, there is some minor corrections and information are necessary:

1. Increase the size of Figure 1A

2. Revise the X and Y-axis numbers of Figure 5. Give some space between the number and the line

3. Give a space between the number and the unit. Example: 86.3 dB 

4. Can you provide an example of a step-by-step signal processed (input signal and output signal) after passing a certain block in the block diagram of Figure 1a? Also add some references when explaining this.

Author Response

Dear reviewer

Thank you for giving us the opportunity to revise the manuscript and many thanks for your comments. All comments have been replied in the attachment, please see the attachment. Thanks.

Best regards

Dr. Wang

Reviewer 2 Report

Comments are attached.

Author Response

(The authors gave the same response as above.)
